# Advantage Constrained Proximal Policy Optimization in Multi-Agent Reinforcement Learning

## Abstract

We explore the combination of value-based method and policy gradient in multi-agent reinforcement learning (MARL). In value-based MARL, *Individual-Global-Max* (IGM) principle plays an important role, which maintains the consistency between joint and local action values. At the same time, IGM is difficult to guarantee in multi-agent policy gradient methods due to stochastic exploration and conflicting gradient directions. In this paper, we propose a novel multi-agent policy gradient algorithm called *Advantage Constrained Proximal Policy Optimization* (ACPPO). Based on *multi-agent advantage decomposition lemma*, ACPPO takes an advantage network for each agent to estimate current local state-action advantage. The coefficient of each agent constrains the joint-action advantage according to the consistency of the estimated joint-action advantage and local advantage. Unlike previous policy gradient-based MARL algorithms, ACPPO does not need an extra sampled baseline to reduce variance. We evaluate the proposed methods for continuous matrix game and Multi-Agent MuJoCo tasks. Results show that ACPPO outperforms the baselines such as MAPPO, MADDPG, and HATRPO.

## 1 Introduction

Many complex sequential decision-making problems in the real world that involve multiple agents can be described by *Multi-Agent Reinforcement Learning* (MARL) problem, including autonomous vehicles (Chang et al., 2021), logistics management (Wang et al., 2017), and electric transportation (Chen & Wang, 2020). MARL in real-world environments usually requires systems with scalability and distributed execution capabilities (Shao et al., 2017). A centralized controller will suffer from the exponential growth of the action space with the number of agents increasing (Hu et al., 2021). Based on the parameter sharing, decentralized policies can reduce the complexity of tasks and enable scalable structures (Shao et al., 2019b). However, directly deploying single-agent reinforcement learning in MARL suffers from a non-stationary issue. To stabilize the training process, MARL introduces monotonic improvement from trust region methods and *Centralized training and Decentralized Execution* (CTDE) from value-based methods.

Trust region learning has played a major role in recent policy gradient methods (Shao et al., 2019a; Kakade, 2001). *Trust Region Policy Optimization* (TRPO) (Schulman et al., 2015) and *Proximal Policy Optimization* (PPO) (Schulman et al., 2017) methods have achieved outstanding performance in single-agent reinforcement learning. The Key point of effectiveness of the trust region methods is based on the theoretically-justified guarantee of monotonic performance improvement at each step. With a KL divergence constraint, parameters can be updated within a trust region that avoids the gradient being too aggressive. Based on parameter sharing, centralized critic, and PopArt (Hessel et al., 2019), MAPPO (Yu et al., 2021) achieves good performance in multi-agent environments. Unfortunately, the value function of MAPPO is affected by the exploration of other agents, making the convergence of MAPPO unstable. Kuba et al. (2021) introduce the optimal baseline for a more accurate estimate of the state value function in MAPPO. The optimal baseline is based on an estimated hypothesized joint action value, which may introduce potential estimation errors. However, examples (Kuba et al., 2022) show that MAPPO does not guarantee consistent improvement even with correct gradients. To obtain the guarantee of monotonic improvement in MARL, Kuba

et al. (2022) introduces HATRPO. HATRPO implements heterogeneous agents and stochastic update schemes of agent gradient directions to obtain guarantee of monotonic improvement. At the same time, HATRPO does not apply a centralized critic but transmits the updated information of previous agents by compound policy ratio. However, the time complexity of the sequential update method is too high, and it is difficult to scale to the scenarios with a large number of agents.

In this paper, we propose *Advantage Constrained Proximal Policy Optimization* (ACPPO), a novel multi-agent policy gradient method based on the advantage decomposition lemma. ACPPO adopts a policy subset, and each agent updates its policy according to their subset. The policy subset avoids the inefficiency caused by sequential updates and the instability caused by important sampling while ensuring improvement consistency gradient updates. A fictitious joint-action advantage function is estimated by summation of a set of local advantages which is learned by each agent. Based on the current advantage and previous advantage, each agent can estimate the consistency of change between their local advantage and joint advantage of the policy subset. The local advantage of each agent will be scaled to ensure that each local policy gradient update improves the performance of the policy subset. In practice, we propose three variants of ACPPO. *Advantage Constrained Proximal Policy Optimization with Hard Threshold* (ACPPO-D) is the combination of the proposed constraint coefficient and PPO. However, the hard threshold is too sensitive to errors. Therefore, we propose ACPPO with a soft threshold, including *Advantage Constrained Proximal Policy Optimization with Parameter-Sharing* (ACPPO-PS) and *Advantage Constrained Proximal Policy Optimization with Heterogeneous-Agents* (ACPPO-HA). The main contributions of this paper are summarized as follows:

- We present a constraint coefficient to the local advantage, which is estimated by the difference between the local and fictitious joint advantage functions, to ensure the consistent improvement of joint policy.

- We propose policy subset to heterogeneous estimate constraint coefficient to ensure monotonic improvement while avoid inefficiency caused by sequential updates and numerical overflow of important sampling.

- We evaluate ACPPO on benchmarks of Multi-Agent MuJoCo against strong baselines such as HATRPO, MAPPO, and MADDPG. The results show that ACPPO achieves state-of-the-art performance across all tested scenarios and demonstrate that the parameter sharing agent without a centralized mixing network performs well in Multi-Agent MuJoCo environments.

## 2 BACKGROUND AND PRELIMINARIES

### 2.1 COOPERATIVE MARL PROBLEM

A fully cooperative multi-agent problem can be described as a six elements tuple $< \mathcal{N}, \mathcal{S}, \mathcal{A}, \mathcal{P}, r, \gamma >$. The $\mathcal{N} = \{1, ..., n\}$ is the set of agents, $\mathcal{S}$ donates the finite state space, $\mathcal{A} = \prod_{i=1}^{n} \mathcal{A}^i$ is the joint action spaces of all agents, $P : S \times \mathcal{A} \times S$ is the transition probability function, $r : \mathcal{S} \times \mathcal{A} \to \mathbb{R}$ is the reward function, and $\gamma \in [0, 1)$ is the discount factor. At each time step $t \in \mathbb{N}$, the agents takes an action $a_t^i \in \mathcal{A}^i$ at state $s_t \in \mathcal{S}$. The combination of each agent's action can be described as joint action $\boldsymbol{a}_t = (a_t^1, ..., a_t^n) \in \mathcal{A}$, drawn from the joint policy $\boldsymbol{\pi}(\cdot|s_t) = \pi^i(\cdot|s_t)$. Based on the $\mathcal{A}$ and $s_t$, the agents receive reward $r_t = r(s_t, \boldsymbol{a}_t) \in \mathbb{R}$, and move to a state $s_{t+1}$ according to the probability $P(s_{t+1}|(s_t, \boldsymbol{a}_t))$. $\rho_0$ is the distribution of the initial state $s_0$, and the marginal state distribution at time $t$ is denoted by $\rho_{\boldsymbol{\pi}}^t$. The state value function and the state-action value function are defined as follows: $V_{\boldsymbol{\pi}}(s) = \mathbb{E}_{a_{0:\infty} \sim \boldsymbol{\pi}, s_{1:\infty} \sim P}[\sum_{t=0}^{\infty} \gamma^t r_t | s_0 = s]$ and $Q_{\boldsymbol{\pi}}(s, \boldsymbol{a}) = \mathbb{E}_{a_{1:\infty} \sim \boldsymbol{\pi}, s_{1:\infty} \sim P}[\sum_{t=0}^{\infty} \gamma^t r_t | s_0 = s, \boldsymbol{a}_0 = \boldsymbol{a}]$. The advantage function can be described as $A_{\boldsymbol{\pi}}(s, \boldsymbol{a}) = Q_{\boldsymbol{\pi}}(s, \boldsymbol{a}) - V_{\boldsymbol{\pi}}(s)$. The objective of the agents in cooperative problem is to maximise the expected return $J(\boldsymbol{\pi}) = \mathbb{E}_{\boldsymbol{s}_{0:\infty} \sim \rho_{\boldsymbol{\pi}}^{0:\infty}, \boldsymbol{a}_{0:\infty} \sim \boldsymbol{\pi}}[\sum_{t=0}^{\infty} \gamma^t r_t]$. The set of all agents excluding agents $(i_1, ..., i_m)$ is represented by $-(i_1, ..., i_m)$. Their local joint state-action value function are defined as $Q_{\boldsymbol{\pi}}(s, \boldsymbol{a}^{(i_1, ..., i_m)}) = \mathbb{E}_{\boldsymbol{a}^{-(i_1, ..., i_m)} \sim \boldsymbol{\pi}^{-(i_1, ..., i_m)}}[Q_{\boldsymbol{\pi}}]$, which is the expected return for the action $\boldsymbol{a}^{(i_1, ..., i_m)}$ chosen by the set of agents $(i_1, ..., i_m)$. The local advantage function is defined as follows: $A_{\boldsymbol{\pi}}(s, \boldsymbol{a}^{(i_1, ..., i_m)}) = Q_{\boldsymbol{\pi}}(s, \boldsymbol{a}^{(i_1, ..., i_m)}, \boldsymbol{a}^{-(i_1, ..., i_m)}) - Q_{\boldsymbol{\pi}}(s, \boldsymbol{a}^{-(i_1, ..., i_m)})$. In additional, the notations $\overline{Q}, \overline{V}, \overline{\boldsymbol{\pi}} = <\overline{\pi}^1, ..., \overline{\pi}^n>$ are used to represent updated $Q, V, \boldsymbol{\pi} = <\pi^1, ..., \pi^n>$.

## 2.2 TRUST REGION METHODS IN REINFORCEMENT LEARNING

To monotonically improve the performance of the agent at each iteration, TRPO (Schulman et al., 2015) and PPO (Schulman et al., 2017) were proposed in single-agent RL. TRPO can be described by the following theorem.

**Theorem 1.** *(Schulman et al., 2015) Let $D_{KL}^{\max}(\pi, \overline{\pi}) = \max_s D_{KL}(\pi(\cdot|s), \overline{\pi}(\cdot|s))$ and $L_\pi(\overline{\pi}) = J(\pi) + \mathbb{E}_{s \sim \rho_\pi, a \sim \overline{\pi}}[A_\pi(s, a)]$. Then the following bound*

$$J(\overline{\pi}) \geq L_\pi(\overline{\pi}) - C D_{KL}^{\max}(\pi, \overline{\pi}) \tag{1}$$

*holds, where $C = \frac{4\gamma \max_{s,a} |A_\pi(s,a)|}{(1-\gamma)^2}$.*

$L_\pi(\overline{\pi})$ is the estimation of the actual performance of the sampled policy $\overline{\pi}$. When the distance between current policy $\pi$ and a sampled policy $\overline{\pi}$ decreases, the accuracy of $L_\pi(\overline{\pi})$ increases. Based on above theorem, the agent updates its policy within trust region at step $k + 1$ according to

$$\pi_{k+1} = \arg\max_\pi (L_{\pi_k}(\pi) - C D_{KL}^{\max}(\pi_k, \pi)). \tag{2}$$

Within trust region, the above update guarantees a monotonic improvement of the policy. In practice, a parameterized policies $\pi_\theta$ is updated as follows:

$$\theta_{k+1} = \arg\max_\theta L_{\pi_{\theta_k}}(\pi_\theta), \text{ subject to } \mathbb{E}_{s \sim \rho_{\theta_k}}[D_{KL}(\pi_{\theta_k}, \pi_\theta)] \leq \delta. \tag{3}$$

At each iteration $k$, TRPO uses heuristic algorithm to search target policy $\pi_{k+1}$ within trust region. To reduce the cost on $\mathbb{E}_{s \sim \rho_{\theta_k}}[D_{KL}(\pi_{\theta_k}, \pi_\theta)]$ when updating $\pi_{\theta_k}$, PPO was proposed by Schulman et al., which uses only first-order derivatives. PPO optimizes the parameters of policy by maximizing PPO-clip surrogate objective

$$L_{\pi_{\theta_k}}^{\mathrm{PPO}}(\pi_\theta) = \mathbb{E}_{a \sim \pi_{\theta_k}, s \sim \rho_{\pi_k}} \min\left[\frac{\pi_\theta(a|s)}{\pi_{\theta_k}(a|s)} A_{\pi_{\theta_k}}(s, a), \mathrm{clip}(\frac{\pi_\theta(a|s)}{\pi_{\theta_k}(a|s)}, 1 \pm \epsilon) A_{\pi_{\theta_k}}(s, a)\right]. \tag{4}$$

The ratio $\frac{\pi_\theta(a|s)}{\pi_{\theta_k}(a|s)}$ beyond the threshold interval $\epsilon$ is clipped to constraint the size of policy updates.

## 2.3 VALUE DECOMPOSITION METHODS

CTDE is a popular cooperative MARL framework. The agent is trained by a centralized critic who has access to the global state and the actions of other agents. IGM principle requests consistency between joint and local greedy action values:

$$\arg\max_{\boldsymbol{a}} Q_{\boldsymbol{\pi}}(s, \boldsymbol{a}) = \left(\arg\max_{a^1} Q(s, a^1), \cdots, \arg\max_{a^n} Q(s, a^n)\right). \tag{5}$$

To implement the IGM principle in value-based multi-agent reinforcement learning, QMIX proposed non-negative parameters mixing network. The mixing network ensure the monotonic relationship between a joint-action $Q_{\boldsymbol{\pi}}(s, \boldsymbol{a})$ and each $Q_\pi(s, a^i)$:

$$\frac{\partial Q_{\boldsymbol{\pi}}(s, \boldsymbol{a})}{\partial Q_\pi(s, a^i)} \geq 0, \forall a^i \in \mathcal{A}^i. \tag{6}$$

## 2.4 TRUST REGION METHODS IN MULTI-AGENT REINFORCEMENT LEARNING

In recent years, a series of trust-region methods were proposed in multi-agent reinforcement learning. A typical parameter sharing approach is MAPPO, where the policy parameter $\theta$ considers the trajectories collected by all agents and is updated by maximizing the objective of

$$L_{\boldsymbol{\pi}_{\theta_k}}^{\mathrm{MAPPO}}(\pi_\theta) = \sum_{i=1}^n \mathbb{E}_{a^i \sim \pi_{\theta_k}, s \sim \rho_{\pi_k}} \min\left[\frac{\pi_\theta(a^i|s)}{\pi_{\theta_k}(a^i|s)} A_{\pi_{\theta_k}}(s, a^i), \mathrm{clip}(\frac{\pi_\theta(a^i|s)}{\pi_{\theta_k}(a^i|s)}, 1 \pm \epsilon) A_{\pi_{\theta_k}}(s, a^i)\right]. \tag{7}$$

HATRPO (Kuba et al., 2022) is a heterogeneous-agent TRPO method based on the multi-agent advantage decomposition lemma, which shows that the joint-action advantage function can be decomposed into a summation of each agent's local advantages.

**Lemma 1** (Multi-Agent Advantage Decomposition). *(Kuba et al., 2022) Given a joint policy $\boldsymbol{\pi}$, for any state $s$, the following equation holds for any subset of $i_{1:m}$ agents in Markov games,*

$$A_{\boldsymbol{\pi}^{i_{1:m}}}(s, \boldsymbol{a}^{i_{1:m}}) = \sum_{j=1}^{m} A_{\boldsymbol{\pi}^{i_j}}(s, \boldsymbol{a}^{i_{1:j-1}}, a^{i_j}). \tag{8}$$

Based on multi-agent advantage decomposition lemma, HATRPO generalizes Theorem 1 to multi-agent systems as follows:

$$J(\overline{\boldsymbol{\pi}}) \geq J(\boldsymbol{\pi}) + \sum_{m=1}^{n} [L_{\boldsymbol{\pi}}^{i_{1:m}}(\overline{\boldsymbol{\pi}}^{i_{1:m-1}}, \overline{\pi}^{i_m}) - C D_{KL}^{\max}(\pi^{i_m}, \overline{\pi}^{i_m})]. \tag{9}$$

The above equation provides the key idea of HATRPO, which is the sequential update scheme. Each policy can guarantee incremental updates. In practice settings, parameterized multi-agent policies $i_m \in \pi_\theta^{i_{1:n}}$ is updated as follows:

$$\theta_{k+1} = \Big(\frac{\pi_\theta^{i_m}(a^{i_m}|s)}{\pi_{\theta_k}^{i_m}(a^{i_m}|s)} - 1\Big) M^{i_{1:m}}(s, \boldsymbol{a}), \text{ where } M^{i_{1:m}} = \frac{\overline{\boldsymbol{\pi}}^{i_{1:m-1}}(\boldsymbol{a}^{i_{1:m-1}}|s)}{\boldsymbol{\pi}^{i_{1:m-1}}(\boldsymbol{a}^{i_{1:m-1}}|s)} \hat{A}(s, \boldsymbol{a})$$

$$\text{subject to } \mathbb{E}_{s \sim \rho_{\boldsymbol{\theta}_{\boldsymbol{\pi}_k}}}[D_{KL}(\pi_{\theta_k}^{i_m}(\cdot|s), \pi_\theta^{i_m}(\cdot|s))] \leq \delta, \tag{10}$$

where $\pi^{i_m}(\cdot|s)$ is designating policies for each state, $\hat{A}(s, \boldsymbol{a})$ is an estimate of the advantage function. To reduce the computation cost of Hessian of the expected KL divergence, HATRPO can be simplified to HAPPO by considering using first-order derivatives as follows

$$\mathbb{E}_{\boldsymbol{a} \sim \boldsymbol{\pi}_{\boldsymbol{\theta}_k}, s \sim \rho_{\boldsymbol{\pi}_k}} \Big[ \min\Big(\frac{\pi_\theta^i(a^{i_m}|s)}{\pi_{\theta_k}^{i_m}(a^i|s)} M^{i_{1:m}}(s, \boldsymbol{a}), \text{clip}(\frac{\pi_\theta^i(a^{i_m}|s)}{\pi_{\theta_k}^{i_m}(a^i|s)}, 1 \pm \epsilon) M^{i_{1:m}}(s, \boldsymbol{a})\Big)\Big]. \tag{11}$$

Due to the update schemes, each agent has to wait for the other agents to finish updating. To take full advantage of update schemes, HATRPO introduces heterogeneous parameters that increase the total weight of agents to make it able to express complex joint policies.

## 3 METHOD

In this section, we propose a novel MARL policy gradient method called ACPPO. ACPPO combines the value decomposition method and policy gradient to satisfy monotonic consistency between joint and local advantage. In Subsection 3.2 we present advantage constrained policy gradient, and in Subsection 3.2 we propose practical applications of ACPPO, including ACPPO-D, ACPPO-PS, and ACPPO-HA.

### 3.1 ADVANTAGE CONSTRAINED POLICY GRADIENT

Based on Lemma 1, similar to QMIX, the estimated joint-action advantage function can be represented by the summation of the local advantage function. By imposing a monotonic constraint on the relationship between $A_{\boldsymbol{\pi}}(s, \boldsymbol{a})$ and $A(s, a^i)$, the global $\arg\max$ on joint-action yields the same results as a set of $\arg\max$ individual action as follows(Wang et al., 2021):

$$\arg\max_{\boldsymbol{a}} A_{\boldsymbol{\pi}}(s, \boldsymbol{a}) = \big(\arg\max_{a^1} A(s, a^1), \cdots, \arg\max_{a^n} A(s, a^n)\big). \tag{12}$$

In CTDE method, the decentralized advantage $A(s, a^i)$ satisfying the IGM principle is estimated by centralized mixing network. If all agents satisfy the IGM principle, the non-empty subset of $\mathcal{N}$ also satisfies the IGM principle. Therefore, the advantage function of the current subset can be estimated as follows:

$$A_{\boldsymbol{\pi}^{i_{1:m}}}(s, \boldsymbol{a}^{i_{1:m}}) = \sum_{j=1}^{m} \alpha_{i_j} A_{\boldsymbol{\pi}^{i_j}}(s, a^{i_j}) \tag{13}$$

Without an unbiased centralized critic, it is difficult to get the accurate $\alpha_{i_j}$. At the same time, the local policy gradient based on decomposition advantage does not guarantee the monotonic improvement of joint policy. To guarantee the monotonic improvement of the joint policy, we evaluate the consistency of the current local and global gradients based on the updated joint policy performance.

Given a designating policies $\tilde{\boldsymbol{\pi}}$, its sampled action can be expressed as $\boldsymbol{a}$. According to Theorem 1, the advantage of policies $\tilde{\boldsymbol{\pi}}$ can be described by $A(s, a^{i_m})$ when the distance between policies $\tilde{\boldsymbol{\pi}}$ and current policies $\boldsymbol{\pi}$ is small. The correlation of the local advantage $A_{\tilde{\pi}^{i_m}}$ to the change of joint-action advantage $A_{\tilde{\boldsymbol{\pi}}^{i_{1:m}}}$ can be described as follows:

$$
\begin{aligned}
\frac{\partial Q_{\tilde{\boldsymbol{\pi}}^{i_{1:m}}}\left(s, \boldsymbol{a}^{i_{1:m}}\right)}{\partial Q_{\tilde{\pi}^{i_m}}\left(s, a^{i_m}\right)} &= \lim_{\tilde{\boldsymbol{\pi}} \to \boldsymbol{\pi}} \frac{Q_{\tilde{\boldsymbol{\pi}}^{i_{1:m}}}\left(s, \boldsymbol{a}^{i_{1:m}}\right) - Q_{\boldsymbol{\pi}^{i_{1:m}}}\left(s, \boldsymbol{a}^{i_{1:m}}\right)}{Q_{\tilde{\pi}}\left(s, a^{i_m}\right) - Q_{\pi}\left(s, a^{i_m}\right)} \\
&= \lim_{\tilde{\boldsymbol{\pi}} \to \boldsymbol{\pi}} \frac{Q_{\tilde{\boldsymbol{\pi}}^{i_{1:m}}}\left(s, \boldsymbol{a}^{i_{1:m}}\right) - V_{\boldsymbol{\pi}}(s) - Q_{\boldsymbol{\pi}^{i_{1:m}}}\left(s, \boldsymbol{a}^{i_{1:m}}\right) + V_{\boldsymbol{\pi}}(s)}{Q_{\tilde{\pi}}\left(s, a^{i_m}\right) - V_{\pi}(s) - Q_{\pi}\left(s, a^{i_m}\right) + V_{\pi}(s)} \\
&= \lim_{\tilde{\boldsymbol{\pi}} \to \boldsymbol{\pi}} \frac{A^e_{\tilde{\boldsymbol{\pi}}^{i_{1:m}}}\left(s, \boldsymbol{a}^{i_{1:m}}\right) - A_{\boldsymbol{\pi}^{i_{1:m}}}\left(s, \boldsymbol{a}^{i_{1:m}}\right)}{A_{\tilde{\pi}}\left(s, a^{i_m}\right) - A_{\pi}\left(s, a^{i_m}\right)} \\
&= \frac{\Delta A^e_{\tilde{\boldsymbol{\pi}}^{i_{1:m}}}\left(s, \boldsymbol{a}^{i_{1:m}}\right)}{\Delta A_{\tilde{\pi}^{i_m}}\left(s, a^{i_m}\right)}
\end{aligned}
\tag{14}
$$

$A^e_{\tilde{\boldsymbol{\pi}}^{i_{1:m}}}\left(s, \boldsymbol{a}^{i_{1:m}}\right)$ is obtained by subtracting the previous $V_{\boldsymbol{\pi}}(s)$ from the current $Q_{\tilde{\boldsymbol{\pi}}}(s, \boldsymbol{a}^{i_{1:m}})$. According to the requirement of consistency between joint advantage and local advantage, the constraint coefficient of the agent $i_j$ is defined as follows:

$$
\alpha_{i_j} = \begin{cases} 1 & \frac{\Delta A^e_{\tilde{\boldsymbol{\pi}}^{i_{1:j}}}\left(s, \boldsymbol{a}^{i_{1:j}}\right)}{\Delta A_{\tilde{\pi}^{i_j}}\left(s, a^{i_j}\right)} > 0 \\ 0 & \frac{\Delta A^e_{\tilde{\boldsymbol{\pi}}^{i_{1:j}}}\left(s, \boldsymbol{a}^{i_{1:j}}\right)}{\Delta A_{\tilde{\pi}^{i_j}}\left(s, a^{i_j}\right)} \leq 0 \end{cases}
\tag{15}
$$

The constrained advantage for updating policy is described as follows:

$$
\overline{A}_{\pi^{i_j}} = \alpha_{i_j} A_{\pi^{i_j}}\left(s, a^{i_j}\right).
\tag{16}
$$

$\alpha_{i_j}$ ensures that only the sampled set of actions that satisfy IGM principle is used to update policy. The decomposition operation avoids sequential updates and the conduction of compound policy ratios. The policies are updated by constrained advantage as follows:

$$
L^{\text{ACPG}}_{\boldsymbol{\pi}_{\theta_k}}(\boldsymbol{\pi}_{\theta}) = \mathbb{E}_{\boldsymbol{a} \sim \boldsymbol{\pi}_{\theta_k}, s \sim \rho_{\boldsymbol{\pi}_k}} \pi^{i_j}_{\theta_k}(a^i|s) \overline{A}_{\pi^{i_j}_{\theta_k}}\left(s, a^{i_j}\right).
\tag{17}
$$

The structure of ACPG introduces value decomposition without centralized critic. This makes it more suitable for MARL tasks with large numbers that exceed the capabilities of centralized critics.

### 3.2 ADVANTAGE CONSTRAINED PROXIMAL POLICY OPTIMIZATION

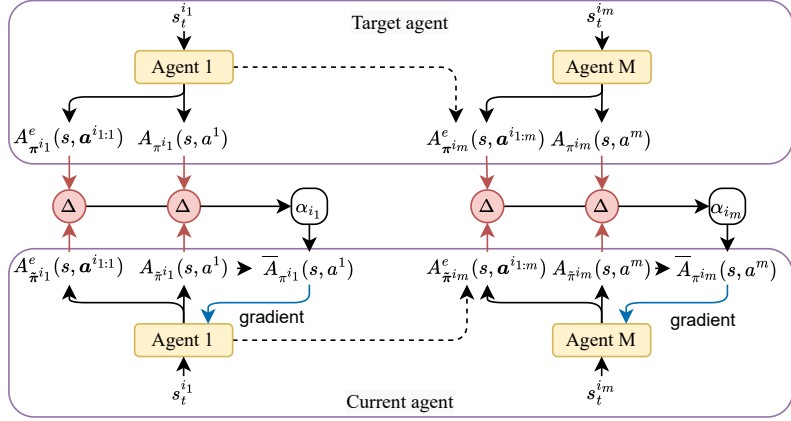

Figure 1: The ACPPO setup. The current network is used to sample data and estimate local advantage. The target network is used to estimate previous advantage.

As shown in Figure 1, to apply equation 17 in practice, we use a actor-critic framework with an additional local advantage function in ACPPO. A target network is introduced to ACPPO to record

the current network for precise estimating KL divergence and previous advantage. At each step, the current advantage function of each agent is optimized by gradient descent to minimize the following loss

$$L^{\text{A}} = \mathbb{E}_{j \in n}(A_{\tilde{\pi}^{i_j}}(s, a^{i_j}) - \hat{A}_{\tilde{\pi}^{i_j}}(s, a^{i_j}))^2, \tag{18}$$

where $\hat{A}_{\tilde{\pi}^{i_j}}(s, a^{i_j})$ could be any kind of estimated advantage, such as GAE. According to $A_{\tilde{\pi}^{i_j}}(s, a^{i_j})$, a fictitious joint-action advantage function $A^e_{\tilde{\boldsymbol{\pi}}^{i_{1:m}}}(s, \boldsymbol{a}^{i_{1:m}})$ could be estimated by summation of the local advantage function as follow:

$$A^e_{\tilde{\boldsymbol{\pi}}^{i_{1:m}}}(s, \boldsymbol{a}^{i_{1:m}}) = \sum_{j=1}^{m} A_{\tilde{\pi}^{i_j}}(s, a^{i_j}). \tag{19}$$

Based on the requirements of cooperative task, a fictitious summation-based joint-action advantage function can represent the trend of joint policy performance. At each epoch, the target network provides the previous local advantage of current sampled action, and is used to calculate the target joint-action advantage. According to the current advantage function and the target advantage function, the constraint coefficient $\alpha_{i_j}$ can be estimated as equation 15. Since it is a hard-threshold variant, we record it as ACPPO-D.

In the early stages of training, the advantage function is inaccurate. Inaccurate estimation may prevent the agent from updating, resulting in too few agents to be updated. Therefore, we introduce a soft threshold that allows more agents to update while maintaining consistency as follows:

$$\alpha^e_{i_j} = \text{clip}(\exp(\frac{A^e_{\tilde{\boldsymbol{\pi}}^{i_{1:j}}}(s, \boldsymbol{a}^{i_{1:j}}) - A^e_{\boldsymbol{\pi}^{i_{1:j}}}(s, \boldsymbol{a}^{i_{1:j}})}{A^e_{\tilde{\pi}^{i_j}}(s, a^{i_j}) - A^e_{\pi^{i_j}}(s, a^{i_j})}), 1 \pm \epsilon). \tag{20}$$

We introduce exponential functions and clip operation to get $\alpha^e_{i_j}$. The exponential function relaxes IGM principles to allow more agents to be updated. The clip operation prevent $\alpha^e_{i_j}$ from overflow or underflow. Bringing the advantage $A_{\boldsymbol{\pi}}(s, \boldsymbol{a})$ and $\alpha^e_{i_j}$ into equation 16 yields the constrained advantage. The policy network of ACPPO is trained by PPO as follows:

$$L^{\text{ACPPO}}_{\boldsymbol{\pi}_{\theta_k}}(\pi_\theta) = \sum_{i=1}^{n} \mathbb{E}_{a^i \sim \pi_{\theta_k}, s \sim \rho_{\pi_k}} \min\left[\frac{\pi_\theta(a^i|s)}{\pi_{\theta_k}(a^i|s)}\overline{A}_{\pi_{\theta_k}}(s, a^i), \text{clip}(\frac{\pi_\theta(a^i|s)}{\pi_{\theta_k}(a^i|s)}, 1 \pm \epsilon)\overline{A}_{\pi_{\theta_k}}(s, a^i)\right],$$
$$\text{where } \overline{A}_{\pi_{\theta_k}}(s, a^i) = \alpha^e_{i_j} A_{\pi_{\theta_k}}(s, a^i). \tag{21}$$

The critic network is optimized to minimize the following loss:

$$L^{\text{V}} = \mathbb{E}_{j \in n}\left[\mathbb{E}_{a_{0:\infty} \sim \boldsymbol{\pi}, s_{1:\infty} \sim P}[\sum_{t=0}^{\infty} \gamma^t r_t | s_0 = s] - V_{\pi_{i_j}}(s)\right]^2. \tag{22}$$

According to whether the method of parameter sharing is adopted, ACPPO is divided into parameter sharing variant called ACPPO-PS and heterogeneous parameters variant called ACPPO-HA. To summarize, Algorithm 1 is proposed as follows:

---

**Algorithm 1** Advantage Constrained Proximal Policy Optimization

---

1: Initialize the joint policy $\boldsymbol{\pi}_0 = (\pi_0^1, ..., \pi_0^n)$.
2: **for** $k = 0, 1, ...$ **do**
3:     Collect sets of trajectories $(s_t, \boldsymbol{a}_t, s_{t+1}, \boldsymbol{r}_t)$ by running policy in the environment.
4:     Backup current network with target network.
5:     Update current advantage network by minimizing loss $L^A$.
6:     **If** soft threshold is true
7:         Compute $\alpha^e_{i_j}$ based on equation (20).
8:     **If** hard threshold is true
9:         Compute $\alpha_{i_j}$ based on equation (15).
10:    Update policy network by minimizing $L^{\text{ACPPO}}_{\boldsymbol{\pi}_{\theta_k}}$.
11:    Update critic network by minimizing $L^V$.
12: **end for**

---

## 4 RELATED WORK

There exists a series of methods that extend policy gradient methods into MARL. IPPO (de Witt et al., 2020) implements vanilla PPO in MARL problem, achieves better results on simple MARL tasks. However, the results show that naively applying PPO in MARL can not guarantee convergence. Yu et al. (2021) proposed MAPPO, which shows the hyper-parameter factors are critical to the performance of MAPPO. MAPPO has to maintain a small KL divergence to avoid policy collapse, severely restricting times of searching iteration. Foerster et al. (2018) proposed the counterfactual baseline to improve efficiency in policy gradient learning. Kuba et al. (2021) proposed optimal baselines in MAPPO to reduce variance in gradient. These methods require additional sampling and repeated estimation, which is difficult to ensure accuracy in practice. Kuba et al. (2022) proposed HATRPO that is guaranteed to monotonic improve the performance of policy and gave detailed proof for the first time. However, each agent has to wait for previous agents to finish updating to get the important sampling ratio as an input factor. For tasks with a large number of agents, the cost of optimization time will be unacceptable and the factor of HAPPO will exponentially increase as the number of agents increases.

Value-based reinforcement learning is another classical MARL method. Sunehag et al. (2018) proposed VDN which represents joint action value function as summation of local action value function to achieve *Centralized Training Decentralized Execution*. Based on VDN, (Rashid et al., 2018) proposed QMIX, which introduces a non-negative mixing network to the centralized critic. The mixing network enforces monotonic consistency between the joint and local action values. To further reduce the learning difficulty, QPLEX (Wang et al., 2021) uses a dueling mixing network to decompose the joint action value into a joint-action advantage and a joint state value. These algorithms require a centralized critic and cannot be directly applied for policy gradient.

## 5 EXPERIMENTS AND RESULTS

In this section, we benchmark ACPPO against the baseline algorithms on continuous matrix game and Multi-Agent MuJoCo (Peng et al., 2021), including HATRPO, HAPPO, MAPPO, MADDPG, and IPPO. The scenario of continuous matrix game is proposed by Peng et al. (2021) to demonstrate the convergence performance of ACPPO and MAPPO on cooperative tasks. Multi-Agent MuJoCo tasks require a set of agents to control different joints of the robot to learn an optimal motion as shown in Figure 5a in Appendix A.3. The number of body joints could be quite large in Humanoid and ManyAgentSwimmer, which increases the costs of learning optimal policy. Table 1 in Appendix A.2 lists the common hyper-parameters used for IPPO, MAPPO, ACPPO-PS, ACPPO-HA, HAPPO, and HATRPO. Table 2 in Appendix A.2 lists the different hyper-parameters. The Adam optimizer constrains the update step size by history gradient variance, so it has the characteristics of stability and has been widely used in the previous methods. However, the Adam optimizer does not fully utilize outlier data, which affects the update efficiency in multi-agent reinforcement learning, so ACPPO uses the RMSprop instead.

### 5.1 CONTINUOUS MATRIX GAME RESULTS.

Experiment results are shown in Figure 4b of Appendix A.1. MAPPO fails to converge stably within $200k$, while ACPPO-PS and ACPPO-D converge to the optimal policy.

### 5.2 MULTI-AGENT MUJOCO RESULTS

Figure 2 demonstrates that, in all scenarios, ACPPO outperforms the selected compare methods, including parameter sharing methods and parameter-independent methods. Experiments show that ACPPO can achieve the best performance in Multi-Agent MuJoCo. With a heterogeneous agent structure, ACPPO-HA can further enhance ACPPO optimization efficiency and stability, but the final performance improvement is not significant. When the number of joints is greater than 10, the time cost of a heterogeneous agent structure with a sequential scheme is unacceptable, including Humanoid and ManyAgentSwimmer. We limit the number of epochs of ACPPO-HA in Humanoid and ManyAgentSwimmer, which severely limits its practical performance. In most scenarios, the performance of ACPPO-D and ACPPO-PS are comparable. As the number of agents increases or the

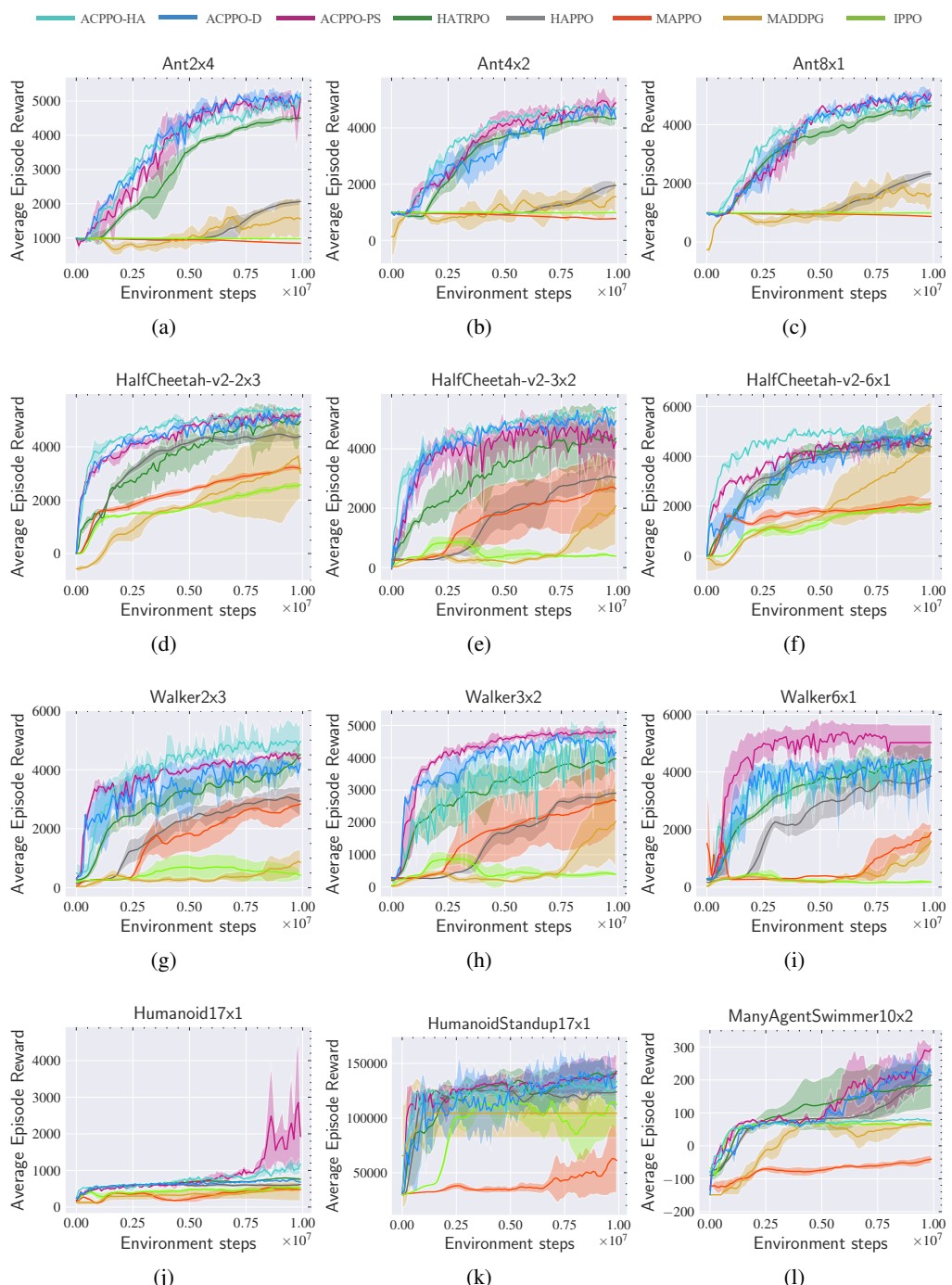

Figure 2: Performance comparison on multiple Multi-Agent MuJoCo tasks. ACPPO-PS and ACPPO-HA outperform selected algorithms in all MuJoCo scenarios. Mean and standard deviation are shown across 3 runs.

task becomes more complex, the variance of the performance of ACPPO-D increases. In Humanoid 17x1, ACPPO-D is worse than HATRPO, indicating that hard threshold truncation is hard to meet the needs of complex tasks.

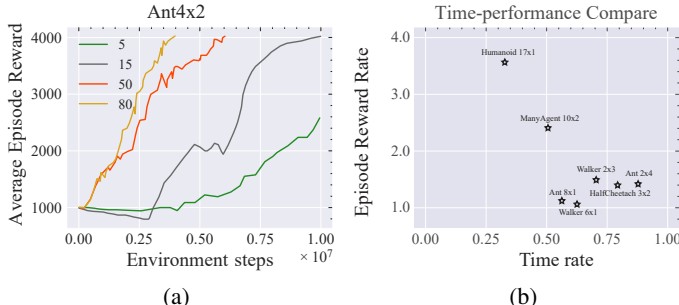

Figure 3: Performance with varying levels of ACPPO-PS policy sample reuse. Time and performance compare rate between HATRPO and ACPPO-PS. The calculation uses the final performance of HATRPO as the denominator. The higher the episode reward rate, the better the performance of ACPPO-PS. The lower the time rate, the faster ACPPO-PS can optimize.

## 5.3 Sample Efficiency

In practice, whether the marginal benefits of independent parameter and sequential update schemes are worth the exponential increase in training time cost is a question worthy of discussion and analysis. First, we evaluate how performing additional epochs during the policy update impacts performance. Figure 3a shows the performance of ACPPO with different numbers of epochs from 5 to 80. As we can see, with more policy epochs, ACPPO can achieve higher data efficiency. It is crucial to improve the performance by increasing the number of epochs to reuse the information of each sampled data. However, due to the potential unstable gradient direction, MAPPO and IPPO do not use a larger number of epochs to avoid policy collapse. Methods with sequential schemes guarantee the monotonic improvement properties of MARL, such as HAPPO and HATRPO. Yet, HAPPO does not use a large number of epochs due to the exponential growth of the compound policy ratio. A growing compound policy ratio could cause overflow to interrupt training. At the same time, HA-TRPO uses the linear programming method to search gradient, and the number of searches is about 200 times. The update step size of ACPPO is constrained by the advantage function and does not use a compound policy ratio, so there is no risk of overflow.

## 5.4 Time Efficiency

We now compare the time costs between sequential updates and synchronous updates. We wonder whether the increase in time cost can bring enough performance improvement. Figure 3b shows the comparison of efficiency and performance between ACPPO and HATRPO. The abscissa is the time comparison. The total training time of HATRPO is used as the denominator, and the total training time of ACPPO is used as the numerator. The ordinate is the performance comparison. The final performance of HATRPO is used as the denominator, and ACPPO is the numerator. As the number of times per agent increases, the overall optimization time increases exponentially in the method with sequential schemes. Therefore, the time cost of the sequential update scheme in multi-agent reinforcement learning is very high.

## 6 Conclusions

In this paper, we propose a novel MARL policy gradient method based on advantage decomposition. The key idea of the algorithm is to introduce constraint coefficients to ensure that the update direction of the local policy is consistent with the joint policy. Based on this, we introduce a practical deep MARL algorithm: ACPPO. Experimental results show that ACPPO achieves state-of-the-art performance in continuous control task Multi-Agent MuJoCo.

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

# A APPENDIX

## A.1 MATRIX GAME

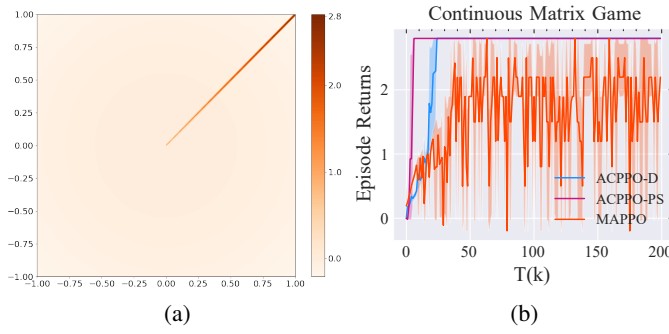

(a)                                    (b)

Figure 4: The continuous matrix game.

Figure 4a shows a simple continuous cooperative matrix game with two agents. One agent can only control the abscissa position, and the other controls the ordinate position. There is a reward function to describe a narrow path from origin to maximum reward point $(1, 1)$. Otherwise, there is a small punishment for moving away from the origin as follow:

$$r(a^1, a^2) = -0.1(x^2 + y^2) + \begin{cases} \frac{10^{-5}}{10^{-5} + |x - y|} & |x - y| < 0.1 \\ & x > 0, y > 0 \\ 0 & \text{otherwise} \end{cases} \quad (23)$$

## A.2 HYPER-PARAMETERS OF ACPPO

Table 1: Common hyper-parameters used for IPPO, MAPPO, HAPPO, HATRPO, ACPPO-PS, and ACPPO-HA in Multi-Agent MuJoCo tasks

| hyper-parameters | value | hyper-parameters | value |
|---|---|---|---|
| critic lr | 5e-3 | std x coef | 1 |
| activation | ReLU | std y coef | 0.5 |
| gamma | 0.99 | max grad norm | 10 |
| hidden layer | 1 | network | mlp |
| hidden layer dim | 64 | num mini-batch | 1 |
| batch size | 4000 | training threads | 8 |
| rollout threads | 4 | episode length | 1000 |
| entropy coefficient | 0.01 | eval episode | 32 |

Table 2: Different hyper-parameters used for IPPO, MAPPO, HAPPO, HATRPO, ACPPO, and ACPPO-HA in Multi-Agent MuJoCo tasks

| Algorithms | IPPO
HAPPO | ACPPO
ACPPO-HA
MAPPO | HATRPO |
|---|---|---|---|
| actor lr | 5e-6 | 5e-6 | / |
| ppo epoch | 5 | 50 | / |
| kl-threshold | / | / | 5e-3 |
| ppo-$\epsilon$ | 0.2 | 0.2 | / |
| accept ratio | / | / | 0.5 |
| optimizer | Adam | RMSprop | Adam |

## A.3 MUJOCO ENVIRONMENT

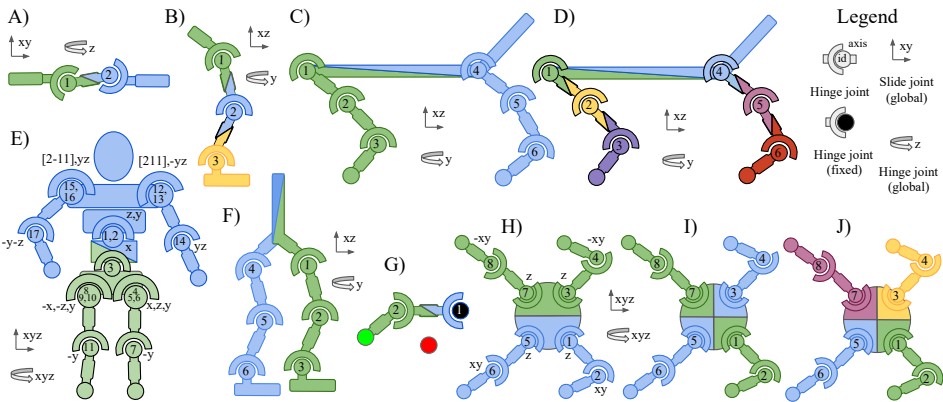

Figure 5: Agent partitionings for Multi-Agent Mujoco environments: A) 2-Agent Swimmer [2x1], B) 3-Agent Hopper [3x1], C) 2-Agent HalfCheetah [2x3], D) 6-agent HalfCheetah [6x1], E) 2-Agent Humanoid and 2-Agent HumanoidStandup (each [1x9,1x8]), F) 2-Agent Walker G) 2-Agent Reacher [2x1], H) 2-Agent Ant [2x4], I) 2-Agent Ant Diag [2x4], J) 4-Agent Ant [4x2]. Colours indicate agent partitionings. Each joint corresponds to a single controllable motor. Split partitions indicate shared body segments. Square brackets indicate [(number of agents) x (joints per agent)]. Joint IDs are in order of definition in the corresponding OpenAI Gym XML asset files (Brockman et al., 2016). Global joints indicate degrees of freedom of the center of mass of the composite robotic agent.

