# OpenReview forum: "Advantage Constrained Proximal Policy Optimization in Multi-Agent Reinforcement Learning"
_ICLR.cc/2023/Conference — Submitted to ICLR 2023_

### Official Review · Reviewer_gE8o · 2022-10-14

**Confidence:** 4
**Correctness:** 2
**Technical Novelty And Significance:** 2
**Empirical Novelty And Significance:** 2
**Recommendation:** 3

**Clarity, Quality, Novelty And Reproducibility:**

- As aforementioned, the method section is hard to understand for me, though I am familiar with the literature.
- Several equations are not supported, making it difficult to judge the mathematical correctness of the proposed method.
- From my current understanding, the novelty of the method is marginal.

**Strength And Weaknesses:**

### About novelty

- This paper seems to be the combination of HAPPO and linear value decomposition. What is the difference between eq. 11 and 21 if we apply the linear value decomposition on the critic of HAPPO? I think the idea of this paper is not quite novel.

### The method section is hard to understand.
- I do not understand how eq. 13 can be reached. What did I miss? The derivation should be given at least in Appendix. Or is it an assumption?
- There are even more issues in eq. 14. Why the partial derivation $\frac{\partial Q_{ \tilde{\boldsymbol{\pi}}^{i_{1:m}}  }}{\partial Q_{ \tilde{\pi}^{i_{1:m}}  }}$ equals to a limit as $\tilde{\boldsymbol{\pi}}$ approaches $\boldsymbol{\pi}$ in the first line? Why the limit in the third line equals to $\frac{\Delta A^e_{ \tilde{\boldsymbol{\pi}}^{i_{1:m}}  } (s,\boldsymbol{a}^{i_{1:m}})   }{\Delta A_{ \tilde{\pi}^{i_{1:m}}  } (s,a^{i_{m}})}$ and what is the definition of $\Delta A^e_{ \tilde{\boldsymbol{\pi}}^{i_{1:m}}  } (s,\boldsymbol{a}^{i_{1:m}})$ and $\Delta A_{ \tilde{\pi}^{i_{1:m}}  } (s,a^{i_{m}})$? Moreover, what is the relation between $\frac{\Delta A^e_{ \tilde{\boldsymbol{\pi}}^{i_{1:m}}  } (s,\boldsymbol{a}^{i_{1:m}})   }{\Delta A_{ \tilde{\pi}^{i_{1:m}}  } (s,a^{i_{m}})}$ and $\alpha_i$ or what is your intuitive idea for the heuristic method in equation (15)?


In this section, many equations, such as 13, 14, 15, and 18, are not well supported, at least not well described. This makes the method hard to understand in its current form. It seems this section should be rewritten.

### More experiments are needed.
- In multi-agent MuJoCo, the proposed method outperforms baselines, but is based on only three seeds, which is not statistically sufficient.
- **SMAC should also be included.**




**Summary Of The Paper:**

The paper proposes a policy optimization method for MARL. To reduce the cost of sequential updates in HATRPO, it uses a network to estimate the local advantage, so agents can be updated efficiently. The consistency between the joint advantage and local advantages is constrained by coefficients, both hard/soft coefficients are explored. Experiments are mainly performed in multi-agent MuJoCo and the proposed method shows better performance than existing work.

**Summary Of The Review:**

It seems the paper is not ready to be published. The method is not sound in its current form, and the method section should be rewritten. Experiments on SMAC should also be included.

---

### Official Review · Reviewer_HtQd · 2022-10-23

**Confidence:** 4
**Correctness:** 3
**Technical Novelty And Significance:** 2
**Empirical Novelty And Significance:** 2
**Recommendation:** 3

**Clarity, Quality, Novelty And Reproducibility:**

The writing of this paper can be significantly improved. The novelty seems insufficient.

**Strength And Weaknesses:**

Strength: The main strength of this paper seems that it proposes a new algorithm for MARL under the CTDE setting. It achieves comparable empirical performances to SOTA methods.

Weakness: It seems that there are some aspects in which this paper could potentially improve.

a.	Novelty. My main concern is about the novelty of this work. The proposed method seems a direct combination of existing methods --- PPO, QMIX, and the decomposition lemma in [Kuba et al, 2022].
b.	Decomposition of Advantage Function. It seems unclear to me the role played by the decomposition lemma. First, in the lemma statement (Lemma 1), what are $ i_{1:m}$? This notation seems not introduced. If this is $i_1, \ldots, i_m$, how do we these agents? Second, what is the advantage of using such a decomposition of advantage? Or where is it used in the algorithm? In (13), the authors construct another set of advantage function by treating the left-hand side as the $Q$ in IGM and finding some factorizations that ensure monotonicity. The left-hand sides of (13) and (8) are the same function, right?
c.	Mathematical rigor. This paper seems to lack some level of mathematical rigor, which makes it hard to understand the merit of the algorithm. For example, when introducing the algorithm in Section 3, the authors handwavingly derive the constraint coefficient $\alpha$ in (14). It is unclear to me how the limit is taken and what $\Delta$ stands for. Is $\Delta$ the derivative? If so, it seems that its computation is both computationally costly and sensitive to errors.
d.	Typos or Bad notation. There seem many typos or bad notation that can potentilly cause confusion. For example, in (12), the $A$ function on the right-hand side should be $A_{\pi}$. In (14), there are $V_{\pi}$ with $\pi$ in boldface font and $\pi$ in a standard font. The indices of agents being $i_1, \ldots, i_m$ are also very confusing.


**Summary Of The Paper:**

This paper studies cooperative multi-agent reinforcement learning under the setting of centralized training and decentralized execution (CTDE). The authors propose a new algorithm based on two algorithm ideas --- PPO and IGM. In particular, PPO specifies how to update the policy in a single-agent RL setting, and IGM yields a decomposition of the Q function that enables the update of the joint policy to be factored into local policy updates in each agent. This paper combines these two approaches, but instead of using the Q function, advantage function is used, which is built on a result about advantage function decomposition proposed in a recent work.

**Summary Of The Review:**

This paper can be improved at least in four aspects: improving the novelty, add better motivation/explanation of the method for estimating the advantage function, more rigorous presentation, and correct typos/notations that cause confusion.

---

### Official Review · Reviewer_ghg2 · 2022-10-24

**Confidence:** 4
**Correctness:** 2
**Technical Novelty And Significance:** 2
**Empirical Novelty And Significance:** 2
**Recommendation:** 3

**Clarity, Quality, Novelty And Reproducibility:**

__Quality__

1. The derivation of the constrained advantage seems to lack theoretical support or sound analysis. How does the authors guarantee that a 0-1 alpha assignment depending on the sign of partial derivatives (Equation 15) satisfies Equation 13, noting that the sign of these derivatives are changing during the learning process.

2. After giving Equation 13, the authors motivate their method by saying that the decomposition cannot guarantee monotonic improvement.  However, although the monotonic improvement is not guaranteed, a linear decomposition guarantees convergence to local optima [Wang et al. 2021, Off-Policy Multi-Agent Decomposed Policy Gradients]. This raises a possible problem: non-monotonic improvement may bring some benefits similar to simulated annealing. Is monotonic improvement a good property in the aspect of increasing the probability of finding global optima?

3. Many claims in the paper are problematic. Some errors occur to very basic knowledge in the MARL field.

> "similar to QMIX, the estimated joint-action advantage function can be represented by the summation of the local advantage function"

In QMIX, the joint-action value function is not a summation of local advantage functions, but a learnable monotonic combination.

> "by imposing a monotonic constraint on the relationship between $A_\pi(s, a)$ and $A(s, a_i)$, the global arg max on joint-action yields the same results as a set of arg max individual action as follows"

The correctness of Equation (12) does not depend on the monotonic constraint. Actually, this holds whenever the IGM holds, which is obvious by subtracting $V$ from $Q$.

**Strength And Weaknesses:**

The reviewer is mainly concerned about the soundness of the proposed method. The use of partial derivatives in masking out action in the calculation of advantage functions is not well supported. In practice, the sign of the derivatives may change abruptly and is vulnerable to gradient noise, which may render the learning unstable.

The background section is clearly written, but the method section has various issues (discussed in the following sections) and is somewhat difficult to follow. The reviewer had to

**Summary Of The Paper:**

The paper studies trust region multi-agent policy gradients. The core contribution is a normalization term when calculating the advantage function. The authors compute the partial derivatives of global Q functions with respect to local utility functions to identify actions subject to the IGM condition. Only those "IGM actions" will be used to calculate advantage functions.

**Summary Of The Review:**

The paper studies an important problems, but there are many issues about the proposed method.

---

### Official Review · Reviewer_U46S · 2022-10-25

**Confidence:** 4
**Correctness:** 3
**Technical Novelty And Significance:** 3
**Empirical Novelty And Significance:** Not applicable
**Recommendation:** 3

**Clarity, Quality, Novelty And Reproducibility:**

Clarity: the paper's exposition is more clearly structured.
Novelty: The article seems to be less innovative and more of an enhancement of experimental results, but its experiments are too homogeneous and insufficient to support its innovation.
Reproducibility:  The author did not provide the source code, I cannot confirm it.

**Strength And Weaknesses:**

Strength:
1. The author presents a constraint coefficient to the local advantage, which is estimated by the difference between the local and fictitious joint advantage functions, to ensure the consistent improvement of a joint policy.
2. The author proposes a policy subset to heterogeneous estimate constraint coefficient to ensure monotonic improvement while avoiding inefficiency caused by sequential updates and numerical overflow of importance sampling.
3. The structure of Figure 1 is more clearly represented.

Weakness:
1. The author seems to have tested only on the matrix game and MAmujoco environments, and as far as I know, there are other environments, including SMAC, Google Football, etc. I think the author's experiments are too single-minded and do not support well the innovation stated in the article.
2. The authors mention several derivatives of ACPPO, including ACPPO-PS, ACPPO-HA, and ACPPO-D, but the descriptions in the text are too omitted, making it very confusing to read. Also based on the experimental results in Figure 2, I could not see any difference between these three algorithms, and also in the Mamujoco environment, there are no distinguishing experimental results.

**Summary Of The Paper:**

The authors propose a novel multi-agent policy gradient algorithm called Advantage Constrained Proximal Policy Optimization (ACPPO).

**Summary Of The Review:**

I don't think this paper deserves to be accepted.

---

### Decision · Program_Chairs · 2023-01-20

**Decision:**

Reject

**Justification For Why Not Higher Score:**

The lack of clarity and possible technical issues make the paper not ready for publication.

**Justification For Why Not Lower Score:**

N/A

**Metareview: Summary, Strengths And Weaknesses:**

Summary:
This paper studies the cooperative setting for MARL under the centralized training and decentralized execution (CTDE) setting. The main idea is to use IGM to decopose the Q-function such that each agent updates its own part locally. The update procedure in each agent is PPO on the advantange function, building on a recent results on advantage function decomposition.

Strength:
- The main strength of this paper seems that it proposes a new algorithm for MARL under the CTDE setting. It achieves comparable empirical performances to SOTA methods.

Weakness:
- Typos and presentations issues make the paper hard to understand.
- Reviewers are concerned about the soundness of the approach due to the lack of theoretical analysis.
- Experimental environment might not be diverse enough
- Authors did not respond to reviewers' concerns.